# Effects of Non-Saccharomyces Yeasts and Their Pairwise Combinations in Co-Fermentation with *Saccharomyces cerevisiae* on the Quality of Chunjian Citrus Wine

**DOI:** 10.3390/molecules29051028

**Published:** 2024-02-27

**Authors:** Yu Fu, Yueyue Gao, Ming Yang, Juan Chen, Chenglin Zhu, Junni Tang, Lianhong Chen, Zijian Cai

**Affiliations:** 1College of Food Science and Technology, Southwest Minzu University, Chengdu 610225, China; yufu@swun.edu.cn (Y.F.); gyy_rita0312@126.com (Y.G.); chenj1221@126.com (J.C.); chenglin.zhu@swun.edu.cn (C.Z.); junneytang@swun.edu.cn (J.T.); lianhong_chen@163.com (L.C.); 2Sichuan Sports College Rehabilitation Research Center, Chengdu 610093, China; my8206@163.com

**Keywords:** Chunjian citrus wine, mixed fermentation, non-*Saccharomyces cerevisiae*, HS-SPME-GC-MS, sensory evaluation

## Abstract

Non-Saccharomyces (NSc) yeasts have great potential in improving wine qualities. In this study, two NSc and two *Saccharomyces cerevisiae* (Sc) samples were tested on their performance of mono-inoculated and composite culture in the fermentation of Chunjian citrus wine. The cell count, Brix degree, total sugar, total acidity, alcohol level, pH value, color intensity (CI), and tonality were determined to evaluate the contribution of NSc to the quality of citrus wine in the mixed fermentation. Volatile compounds were analyzed by HS-SPME-GC-MS, and sensory evaluation was carried out. During the 9-day fermentation, the mixed-culture wine exhibited a higher cell concentration than the pure culture. After the fermentation, mixed-culture wine specifically decreased the concentrations of unfavorable volatile compounds, such as isobutanol and octanoic acid, and increased favorable volatile compounds, including ethyl octanoate, ethyl decanoate, and phenylethyl acetate. The quality category of the citrus wine was improved compared with the Sc mono-inoculated wines, mainly in regard to aroma, retention, and sweetness. The study shows that the mixed fermentation of NSc and Sc has positive impacts on reducing alcohol level and total acidity and increasing CI. The present work demonstrates that the mixed fermentation of NSc and Sc has enormous beneficial impacts on improving the quality of citrus wine.

## 1. Introduction

The sensory profiles of wines produced by monoculture differ substantially from those that are diverse in microbial communities [1]. Meanwhile, there is a growing interest in studying and characterizing non-*Saccharomyces* (NSc) yeasts for the development of starter cultures [2]. Studies have shown that some NSc yeasts can contribute to a more enriched aroma and flavor of wine products, owing to their unique metabolites produced under the conditions of the early stage of fermentation. The interaction of NSc and *Saccharomyces* (Sc) yeasts in the mixed fermentation process can affect the produced amount of ethanol, pH value, and the production of volatile aroma components, especially alcohols and ester compounds [3,4,5]. Nsc usually plays an important role at the beginning of the fermentation; however, most of them cannot complete the fermentation [6]. Therefore, NSc yeasts are used in pairs with the Sc yeast wine strain, which has great potential for not only optimizing the fermentation process but also improving the quality of fruit wine by enriching the aroma profile [7,8].

Citrus fruits are among the most important fruits widely distributed around the world [9]. As one of the most consumed fruits, they also have great economic importance [10]. Citrus fruits, such as orange, mandarin, and grapefruit, are used for wine and spirit production [11], forming varieties of citrus wines, including orange wine, lemon wine, mandarin wine, and blends (blended with other fruits or grapes). Citrus wines come in a spectrum of hues from light gold to red, but a clear light amber with no sediment is generally preferred [12]. Citrus wine is widely consumed in the United States, Europe, and Asia, and the market share is growing rapidly globally [13]. Although citrus wine is not as familiar as grape wine to customers, the production of citrus wine has a long history in the Mediterranean and Asia, such as Italy’s Limoncello, Spain’s Arancello, and Japan’s yuzu [11,14,15]. Meanwhile, the large production of citrus fruits, short harvest time, and short shelf life have led to the development of citrus wine production in China [16]. Citrus reticulata Chun Jian, also known as Baba citrus, is a Japanese hybrid variety of Okitsu No. 44 (*C. reticulata* × *C. sinensis*) [17]. It has gained great popularity due to its flavorful, juicy, seedless, and easy-to-peel flesh. However, early ripening in Chunjian citrus after a high yield often occurs in piled-up stock, leading to rot, waste, and environmental pollution [16]. One solution to the problem of unsold Chunjian citrus is to produce fruit wine, which has great potential for new product development for the wine industry.

To explore the potential of NSc and Sc yeast co-fermentation for citrus wines, fermentations with Sc monocultures and Sc-NSc co-cultures were compared. In this study, physical and chemical indicators and a sensory evaluation of the fermented citrus wines with mixed and monoculture-inoculated fermentations were determined, and the volatile compounds were characterized using HS-SPME-GC-MS.

## 2. Results

### 2.1. Growth and Survival of NScs and Scs in Pure and Mixed Cultures

Yeast population evolution in the mono-culture fermentation process is shown in Figure 1a. Mp and Ka exhibited similar growth patterns, both ending the logarithmic growth phase on day 6 with a maximum biomass population of 10^8^ CFU/mL (Figure 1a). F15 and D254 ended their logarithmic growth phase on day 5 with a maximum biomass population of 5.6 × 10^7^ CFU/mL. Throughout the fermentation process, Sc yeasts showed a faster reproduction rate than NSc yeasts. Additionally, D254 demonstrated a higher cell concentration (6.3 × 10^7^ cells/mL) compared to F15 (2.0 × 10^7^ cells/mL) at its peak, indicating that D254 yeast has better tolerance and greater fermentation potential than F15. In the later stage of fermentation, the total number of cells of the four types of yeast showed a gradual downward trend after consuming air, nutrients, and living space on day 9 (Figure 1a).

In the mixed fermentation system, Sc and NSc were inoculated in a 1:10 ratio (Figure 1b). In this study, Sc quickly adapted to the pre-existing NSc environment and maintained a relatively high growth rate within 4 days. Among all mixed-fermentation groups, Mp-F15 grew the fastest during the early stage of the co-culture, reaching a density of 5.6 × 10^8^ CFU/mL on day 6, one day later than the other groups. This was followed by Ka-F15 peaking at 3.2 × 10^8^ CFU/mL on day 5; this was significantly higher than when they were monocultures.

### 2.2. Consumption of Sugar in Mono-Culture Fermentation and Mixed Fermentation

By measuring the sugar content change alongside the fermentation period, the efficiency of utilizing sugars in the citrus juices was studied. Mp and Ka had the most sugar remaining after fermentation, with Mp slightly below Ka in final sugar content (6.1 vs. 6.5 °Brix, Figure 2). Compared to NSc, Sc has a much higher efficiency in converting sugars in fruit juice into alcohol (sugar consumption rate: 87% of F15 and D254 vs. 68% of Mp and Ka). The mixed-culture groups consumed a range of 17.2 to 18 °Brix, substantially exceeding both NSc’s maximum of 13.8 °Brix and Sc’s 17.3 °Brix. The Mp-F15 peaked at 18.1 °Brix, which clearly demonstrated the most efficient sugar utilization.

### 2.3. Physical and Chemical Properties of Citrus Wine

As shown in Figure 3a, Mp-F15 and Ka-D254 retained lower total sugar (3.8 g/L) than Mp-D254 (4.25 g/L) and Ka-F15 (4.4 g/L). Notably, mixed-fermented wines maintained a slightly higher sugar level (3.8–4.4 g/L) compared to the NSc pure-fermentation groups (3.54 g/L for F15 and 3.68 g/L for D254). NSc mono-cultures produced significantly lower alcohol compared to both Sc mono-cultures and mixed fermentations. While NSc’s Mp and Ka cultures reached only 4.52% and 3.87%, respectively, Sc’s F15 and D254 achieved 10%. Moreover, mixed fermentations yielded a range of 8.0% to 9.6% (Figure 3b). *M. pulcherrima* reduced alcohol content in co-fermentation by 0.4% (F15) and 1.93% (D254). Meanwhile, the co-fermentation of *K. apiculata* with *S. cerevisiae* reduced alcohol content by 1.73% (F15) and 1.08% (D254).

Additionally, mixed fermentation demonstrated a remarkable ability to decrease the total acid content in citrus wines (Figure 3c). For example, Mp and Ka co-fermented with F15 (Mp-F15 and Ka-F15) had significantly lower total acidity than F15 alone (5.69 vs. 5.63 vs. 7.44 g/L). Similarly, Mp and Ka co-fermented with D254 (Mp-D254 and Ka-D254) also decreased the total acidity compared to D254 alone (5.88 vs. 6.00 vs. 7.31 g/L), despite minimal variations in pH value (Figure 3d).

Color measurements of the different citrus wine were conducted (Table 1). Mp-F15 mixed-fermentation wine showed significantly higher values for CI (average value of CI = 1.44), followed by Ka-D254 (CI = 0.88), with an intensified color in the fermentation compared to the pure culture. Moreover, color tonality (CT) was significantly lower for the assays of mixed fermentation than Sc mono-culture fermentation. For example, Co-fermenting Mp and Ka with either F15 (Mp-F15 and Ka-F15) or D254 (Mp-D254 and Ka-D254) significantly decreased CT compared to their respective monocultures (e.g., Mp-F15: 1.54 vs. F15: 4.16; Ka-D254: 2.90 vs. D254: 4.85). In this study, notable evidence of the effect of NSc on the color of citrus wine was observed. The Sc and NSc mixed fermentations had a yellower, redder color, presenting an amber color in the wine, while the Sc pure-cultured wine had a more bluish and light yellowish color, indicating less matured wine properties.

### 2.4. Volatile Compounds in Citrus Wine

A total of 40 volatile components were detected by GC-MS analysis, including 11 alcohols, 12 esters, 5 acids and ketones, 2 terpenes, and 5 other compounds such as phenols, aldehydes, and olefins (Figure 4a,b). Twelve aroma compounds were present in all samples, including 1-pentanol, isobutanol, 3-methyl-1,5-pentanediol, ethyl acetate, phenylethyl acetate, ethyl phenylacetate, acetic acid, octanoic acid, isovaleric acid, cyclohexanone, and 2,4-di-tert-butylphenol (Figure 4b). Overall, mixed-fermentation wines were more diverse in aroma components than Sc mono-inoculated fermentation, indicating that mixed fermentation can have a positive effect on the flavor diversity of Chunjian citrus wine.

Seven to eleven higher alcohols were detected in the eight Chunjian citrus fermented wines, with the highest content in the Mp-F15 group, which can increase the complexity of wine aroma. In this study, the addition of NSc yeasts in the fermentation inhibited the production of isobutanol compared with Sc pure culture (Figure 4b). *S. cerevisiae* cultures (F15: 990 μg/L, D254: 1097 μg/L) produced significantly more isobutanol than *M. pulcherrima* (49 μg/L). This difference persisted in co-fermentations, with Mp-F15 and Mp-D254 showing lower isobutanol levels (942 μg/L and 753 μg/L, respectively). Moreover, co-fermenting F-15 with non-*Saccharomyces* yeasts increased phenylethanol production. Compared to the pure-culture level of 198 μg/L, the presence of *M. pulcherrima* and *K. apiculata* led to increases of 36% (269 μg/L) and 94% (385 μg/L), respectively.

Esters are the most enriched group of volatile compounds in citrus wines (Figure 4b). Mixed-fermentation group Mp-F15 contained the most ester substances, reaching a level of 6 g/L. The total amounts of esters detected in the analyzed citrus wines were 4.6 mg L^−1^ for NSc pure cultures and 3.2–3.8 mg L^−1^ for Sc pure cultures, while *M. pulcherrima* increased ester concentration by 38% for F15 (Mp-F15: 6.2 mg L^−1^) and 41% for D254 (Mp-D254: 5.6 mg L^−1^), and *K. apiculata* contributed a 14% (Ka-F15: 4.4 mg L^−1^) and 41% (Ka-D254: 5.4 mg L^−1^) increase.

According to the ROAV value, ethyl acetate, ethyl octanoate, and ethyl phenylacetate constitute the fundamental flavor components of citrus wine; among them, ethyl acetate has the largest proportion of ester substances in Chunjian citrus wine (ROAV > 1). The ester level of the mixed-fermentation group was significantly increased compared to the Sc mono-inoculated fermentation groups, indicating that NSc produces a large amount of esters, enhancing the flavor of Chuanjian citrus wine (Figure 4b). Among them, ethyl octanoate (pineapple, pear aroma), ethyl acetate (fruit aroma, flower aroma), and ethyl phenylacetate (frankincense, cream) contribute significantly to the aroma of citrus wines (ROAV > 1). In addition, ethyl caproate (melon aroma) and phenylethyl acetate (green apple aroma) contribute the fruity flavor to citrus wines (ROAV > 1).

A total of five acids were detected. Interestingly, octanoic acid, instead of acetic acid, reached the highest concentration amongst all acids in citrus wine. However, the octanoic acid concentration in the mixed-fermentation groups was significantly lower compared to the mono-cultured wine (ROAV > 0.1, Figure 4), which could bring a sour cheese odor to the wine. Overall, the Mp-F15 group had a reduced acid profile compared to the other mixed-fermentation groups (Figure 4b), indicating that mixed fermentation can significantly reduce the total amount of acids, thereby reducing the adverse effects in Chunjian citrus wine.

The Ka-F15 group had the most ketone substances, reaching a concentration of around 189 μg/L, contributing to the aroma of Chunjian citrus wine (Figure 4b, ROAV > 0.1). *M. pulcherrima* and *K. apiculata* produced linalool and citronellol concentrations of 125 μg/L and 173 μg/L, respectively. All the citrus wine samples in this study contained linalool with an ROAV value above 1, which could significantly increase the floral aroma of citrus wine. In addition, the NSc pure-fermentation group presented a higher content of citronellol, which was undetectable in the Sc pure-fermentation group (Figure 4b). In NSc and Sc mixed fermentation, citronellol provided a rose fragrance for the citrus wine and the total concentration of terpene was increased compared with the Sc pure culture. Volatile compounds such as 2,4-di-tert-butylphenol, 2-vinylaldehyde, and D-limonene were also detected in the citrus wine (ROAV > 1), imparting a grapefruit aroma to the Chunjian citrus wine.

To identify the volatile compounds that differentiate the aroma profiles of pure-cultured and mixed-fermentation citrus wines, a supervised PLS-DA model was established. As shown in Figure 5a, clear separation among the groups was observed, especially between the mixed-fermentation groups and pure-cultured groups (Figure 5a). Based on the VIP calculated by the PLS-DA model, ethyl octanoate, ethyl acetate, 3-methy-1-5-pentanediol, phenylethyl acetate, α-Terpineol, Diethyl succinate, 3-hexanol, and ethyl phenylacetate were characteristic flavor compounds that exhibited significantly higher concentrations in mixed-fermented citrus wines than pure-cultured ones (Figure 5b, VIP > 1). Meanwhile, octanoic acid was one of the unfavorable compounds that significantly distinguished between pure-cultured wine and mixed-fermentation wine (Figure 5b, VIP > 1).

### 2.5. Sensory Evaluation

NSc mono-inoculated fermented wines generally scored higher in aroma retention, aroma, and sweetness than Sc mono-inoculated fermented wines, but lower in smoothness, transparency, and color. The sensory evaluation revealed a significant increase in color preference for Mp-D254 and Ka-D254 (mean scores 4.2 and 4.1) compared to their pure-fermented counterparts (Mp, Ka, and D254, mean scores 2.1, 3.6, and 3.1, respectively). Similar improvements were observed for Mp-F15 and Ka-F15 (mean scores 3.6 and 3.3) compared to pure F15 (mean score 2.7). Among the mixed-fermentation wines, the wine containing Ka had the highest aroma retention score (aroma retention scores of Ka-F15 and Ka-D254 were 4.0 and 4.2 out of 5, respectively (Figure 6)). In this study, the Mp-F15 group had the lowest sour taste sensory score (0.67 out of 5), indicating that the mixed strain of Mp and F15 is beneficial for reducing the sour taste of Chunjian citrus wine.

## 3. Discussion

Fruit wine alcohol content typically varies between 5 and 13% [18]. Fermentations with solely *M. pulcherrima* or *K. apiculata* strains in this study fell outside this range. *S. cerevisiae* co-inoculation led to both a remarkable increase in sugar consumption and a substantial increase in alcohol content (8–9.6%). This supports previous studies by demonstrating the requirement of *S. cerevisiae* for a complete fermentation process [19] due to the low sugar utilization efficiency of NSc.

Studies have reported that *M. pulcherrima* can reduce the alcohol concentration in wine [20,21], while some reports showed that it did not significantly affect the ethanol content [22,23]. In this study, *M. pulcherrima* reduced the alcohol content in co-fermentation by 0.4% (F15) and 1.93% (D254). Meanwhile, the co-fermentation of *K. apiculata* with *S. cerevisiae* reduced alcohol content by 1.73% (F15) and 1.08% (D254), which is in accordance with a previous study where alcohol concentration was reduced in wine with *K. apiculata* by 1.79% [24]. Driven by consumer demand and a desire to mitigate the impact of rising global temperatures on viticulture, there is a growing interest in reducing alcohol levels in wine [21,25,26,27]. The respiration of sugars by NSc yeasts has emerged as a promising approach for lowering alcohol levels in wine [25]. The non-*Saccharomyces* strains *M. pulcherrima* and *K. apiculata* were unable to ferment the total amount of sugar; co-fermentation of them with Sc yeast can slow sugar consumption, leading to reduced alcohol production, aligning with the growing consumer demand for lower alcohol wines.

Previous research has suggested that the use of NSc yeasts to reduce alcohol concentration in wine can also lead to the production of significant levels of higher alcohols [21]. In particular, isobutanol has been linked to undesirable greenish or vegetal aromas in wines, especially when it co-occurs with isoamyl alcohol [28,29]. The present study found that, while the addition of NSc yeasts increased the overall diversity of higher alcohols produced, it also significantly reduced the production of isobutanol compared to Sc pure-culture fermentation. For example, *M. pulcherrima* was previously reported to be able to synthesize isobutanol and phenylethanol [30,31]; however, in this study, Mp pure culture produced little isobutanol (49 μg L^−1^), far less than F15 (990 μg L^−1^) and D254 (1097 μg L^−1^). Moreover, in the mixed fermentation, Mp reduced the production of isobutanol in the wine (942 and 753 for Mp-F15 and Mp-D254, respectively), which indicated that the biosynthesis of isobutanol is probably strain-dependent. Interestingly, neither *M. pulcherrima* nor *K. apiculata* produced phenylethanol in the pure-culture fermentations, but co-fermenting with Sc strains led to increased phenylethanol production, a key contributor to rose-like aromas in citrus wine. For example, F-15 produced 198 μg/L phenylethanol in pure culture, but this concentration increased to 269 μg/L and 385 μg/L when co-fermented with *M. pulcherrima* and *K. apiculata*, respectively. This suggests that the application of *K. apiculata* and *M. pulcherrima* had a positive impact on the sensory profile of citrus wine by promoting the production of desirable aroma compounds like phenylethanol while potentially inhibiting the formation of undesired compounds like isobutanol. Further research could explore the mechanisms behind these interactions and their potential for optimizing citrus wine aroma profiles.

Ethyl esters such as ethyl octanoate, ethyl decanoate, and phenylethyl acetate are characterized by fruity and floral notes [32,33]. Esters constitute the most enriched group of volatile compounds in Chuanjian citrus wines, and NSc significantly contributes to enhancing the overall ester concentration. The total amount of esters detected in the analyzed citrus wines ranged from 3.3 to 4.6, and from 4.4 to 6.1 mg L^−1^, for pure cultures and mixed cultures, respectively. Co-fermenting with both *M. pulcherrima* and *K. apiculata* alongside *S. cerevisiae* strains F15 and D254 resulted in a significant increase in total ester production compared to *S. cerevisiae* monocultures. Notably, *M. pulcherrima* increased ester concentration by 38% for F15 and 41% for D254, while *K. apiculata* contributed a 14% and 41% increase, respectively. The contribution of NSc remains unclear, as the ester production could be strain-dependent [34], the result of the enzymatic activity of yeasts [35], and depend on the fermentation conditions and the interaction of yeasts in sequential fermentation [36,37].

Additionally, excessive quantities of volatile acids are seen as a spoilage characteristic, conferring the wine an acrid taste and an unpleasant vinegar aroma [38]. Compared to mono-culture fermentation, mixed fermentation resulted in lower levels of octanoic acid and total acidity in the analyzed wine. Notably, co-fermenting with *M. pulcherrima* and *K. apiculata* led to a remarkable reduction in octanoic acid, an unpleasant-smelling fatty acid, in citrus wine. Compared to pure *S. cerevisiae* fermentations with octanoic acid levels ranging from 3.2 to 3.9 mg/L, mixed cultures significantly lowered this compound to between 0.5 and 0.9 mg/L, representing a reduction of 12–23%. While previous reports suggest that *M. pulcherrima* metabolism can contribute to octanoic acid production through aliphatic carboxylic acid pathways [39,40], this study observed no such correlation. This suggests that *M. pulcherrima*’s octanoic acid production might be strain-dependent.

Total acidity plays a vital role in wine sensory perception, and directly influences the overall organoleptic character of wines [41]. Insufficient acidity causes a bitter and monotonous taste, while excessive acidity can lead to improper smells and astringency in wine. The present study showed that the total acids in the mixed-fermentation citrus wine were slightly lower than that in an orange wine reported in a previous study (<6.0 ± 0.16 g L^−1^) [42]. Interestingly, in a previous study of the association of *S. cerevisiae* and *M. pulcherrima*, the detection of a significant decrease in total acidity in the final wines was observed [43], which was consistent with our study. Therefore, the combination of *S. cerevisiae* and *M. pulcherrima* has a positive effect on reducing the total acidity in wine fermentation. This aligns with previous reports on *K. apiculata*’s ability to lower volatile acidity in wine [24].

Ketones belong to carbonyl compounds that are synthesized from the metabolism of unsaturated fatty acids under lipoxygenase activity during the fermentation process and enhance the aroma effects in fruit wines [44,45]. Mixed fermentation with *M. pulcherrima* and *K. apiculata* increased the total ketones in citrus wines by 12–80%. Terpene compounds have a strong aroma and low sensory threshold, making a significant contribution to the aroma component of fruit wine [46]. Both *M. pulcherrima* and *K. apiculata* produced linalool and citronellol in this study, with linalool concentrations of 125 μg/L and 173 μg/L, respectively. This is lower than the 0.32 and 1.44 mg/L rates reported in a previous study using sequential inoculation with *M. pulcherrima* and *S. cerevisiae* for Pecorino white wines [47].

Color is one of the most important attributes in wines [48]. It was proved that the use of some NSc species can be effective in color protection through pH modification or the formation of stable pyran anthocyanins or polymeric pigments [40]. This study demonstrated that mixed fermentation exerted a greater enhancing effect on CI and CT compared to Sc mono-fermentation in Chuanjian citrus wine. The amber color of citrus wine produced through mixed fermentation received significantly higher color preference scores in the sensory tests compared to the light-yellow tone of the Sc pure-fermented wine. This suggests potential consumer preferences for citrus wine hues.

Currently, citrus wine production primarily involves juice extraction, filtration, and fermentation using *S. cerevisiae*. This study unveils the groundbreaking application of non-*Saccharomyces* yeast (*K. apiculate*, *M. pulcherrima*) in citrus wine fermentation. By uncovering its impact on diverse flavor profiles, alcohol production efficiency, color, and sensory attributes (aroma, taste, mouthfeel), this study provides valuable insights for crafting innovative citrus wines and advancing quality-control practices in industrial production.

## 4. Materials and Methods

### 4.1. Sample Processing

Chunjian citrus fruits were collected from Chengdu Chunjian Agricultural Technology Co., Ltd. (Chengdu, China). Twenty-five kilograms of mature and fresh fruits were cleaned and destemmed. The peel and flesh were blended in a 1:5 ratio, and 80 mg/L of SO_2_ (80 mL/L of sulfite solution) was added to the fruit pulps for anti-corrosion treatment. Pectinase was then used for hydrolysis, and the mixture was filtered and pasteurized at 65 °C for 30 min.

The pasteurized citrus pulps were filtered through a 500-mesh sieve and 25 μm filter cloth, and then packed into sterilized fermentation tanks, with a net content of 300 mL per tank. The total acid content was 5.64 g/L, the pH was 4.06, and the Brix degrees were adjusted from 12.5 °Brix to 20 °Brix.

### 4.2. Yeast Preparation

NSc yeasts *K. apiculate* CICC 1466 and *M. pulcherrima* CICC 32343, and SC yeasts *S. cerevisiae* F15 and *S. cerevisiae* D254, purchased from China Center of Industrial Culture Collection (CICC) (China National Research Institute of Food & Fermentation Industries Co., Ltd., Beijing, China), were used for fermentation. *Kloeckera apiculata* CICC 1466 was originally isolated from Bretagne, France. *Saccharomyces cerevisiae* CICC 31964 has been widely used for wine production. *Saccharomyces cerevisiae* CICC 31898 is widely used for white wine production. *Metschnikowia pulcherrima* CICC 32343 is used for fruit wine production. Additional information on the four yeasts can be seen at http://sales.china-cicc.org/ (accessed on 14 January 2024).

To prepare the yeast suspension, the four types of freeze-dried yeast powder were resuspended in sterile water at 35 °C for 30 min. A small amount of the yeast suspension was inoculated in Yeast Extract Peptone Dextrose (YEPD) medium and cultured at 28 °C for 1 day to obtain single yeast colonies. Strains with obvious colony characteristics were selected and inoculated in YEPD broth with shaking at 28 °C and a speed of 120 rpm for 1 day. The suspension was then diluted serially, and the number of cells was determined using a colony counter. A concentration of 10^6^ CFU/mL of yeast was inoculated into the fermentation liquid.

### 4.3. Mono-Culture and Mixed Fermentation

For single-strain fermentation, we sterilely inoculated *K. apiculate* (Ka) and *M. pulcherrima* (Mp) at a concentration of 1 × 10^7^ CFU/mL and *S. cerevisiae* (F15) and *S. cerevisiae* (D254) at a concentration of 1 × 10^6^ CFU/mL into four groups of fermentation tanks, with three replicates of each group. For mixed fermentation, two NSc strains, CICC 1466 and CICC 32343, were separately inoculated into fermentation tanks at a concentration of 10^7^ CFU/mL each. After 24 h, two Sc strains, F15 and D254, were separately inoculated into the former tanks at a concentration of 10^6^ CFU/mL each. This resulted in four fermentation groups: Mp × F15, Mp × D254, Ka × F15, and Ka × D254. Each group had three replicates.

The eight groups of tanks were placed in the dark at 25 °C for further fermentation. The number of yeast colonies and the Brix degrees in the samples were measured daily. Fermentation was stopped when the number of yeast colonies and the reducing sugar content stabilized. The wine was then centrifuged at 4000 rpm for 10 min, and supernatant was then collected and stored at 4 °C for further evaluation.

### 4.4. Determination of Reduced Sugar, Alcohol Level, pH Value, and Total Acid

Alcohol concentration was tested by an alcohol meter (Yida, Hebei, China). The pH values were measured by a pH meter (Leici, Shanghai, China). Total acid content was determined by titrating the total organic acids in the sample with sodium hydroxide to an endpoint of pH 8.2. Reduced sugar content was determined by Fehling’s solution colorimetric method.

### 4.5. CI and CT Determination

Color intensity and the tonality of the wine were determined according to Liu et al. [49]. Light absorbances at 420, 520, and 620 nm were determined by a spectrophotometer (Aoyi, Shanghai, China). CI was recorded as the sum of these three absorbances. CT was recorded by the ratio between absorbances at 420 and 520 nm. The percentages of yellow (%Yellow), red (%Red), and blue (%Blue) elements were expressed as the ratios of absorbances at 420, 520, and 620 nm to color intensity, respectively [50].

### 4.6. Solid-Phase Microextraction–Gas Chromatography–Mass Spectrometry (SPME-GC-MS) Analysis

Volatile compounds were extracted using SPME; 5 mL of each sample was added into a 15 mL headspace bottle with 5 μL 3-octanol as an internal standard. After thoroughly stirring on a magnetic stirrer, volatile compounds were extracted by heating at 40 °C for 60 min. Then, the extraction head (activated at 230 °C for 30 min) was used to adsorb VOCs for 40 min in the headspace. Then, the fiber head was inserted into the GC injection port for 5 min for analysis.

A Thermo Trace DSQ II GC/MS instrument was used to analyze the volatile flavor compounds, equipped with a DB-Wax gas chromatography column (60 mm × 0.25 mm × 0.25 mm). Helium was used as carrier gas with a flow rate of 1 mL min. The injector temperature was maintained at 250 °C with an unsplit inlet. The temperature in the GC oven was maintained at 35 °C for 5 min, and then raised to 110 °C at a rate of 5 °C/min and maintained for 2 min. Then, it was raised to 150 °C at a rate of 6 °C/min and was maintained for 2 min. Afterward, the temperature was raised to 220 °C at a rate of 8 °C/min and was maintained for 10 min.

The ion source was EI and the temperature was set at 200 °C. The interface temperature was set at 220 °C. A positive ionization mode was applied with an electron energy of 70 eV, a detection voltage of 1000 V, and an emission current of 100 μA. The recorded mass spectra were compared with the NIST 14.0 database to calculate the content of volatile compounds. The peak area normalization method was used in a relative quantitative calculation to obtain the relative percentage content of each compound in the sample.

### 4.7. Calculation of ROAV

The relative odor activity value (ROAV), ranging from 0 to 100, was utilized to assess the individual contribution of each compound to the overall aroma profile. It was computed using the following formula:$$\mathrm{ROAVi}=100\times \frac{\mathrm{OAVi}}{\mathrm{OAVmax}}$$
where OAVmax is the highest odor activity value (OAV) among the volatile compounds and OAVi is the OAV of a specific volatile compound. The OAV is calculated using the equation OAVi = Ci/OTi, where Ci is the concentration of the volatile compound in the sample and OTi is the odor threshold in water expressed in ppb. The odor thresholds of the compounds in water were identified from the literature [51,52].

### 4.8. Sensory Evaluation

The sensory panel consisted of six experienced students with no dietary biases or anaphylactic reactions (3 female students and 3 male students, aged 20–22 years old). The eight types of fermented wine produced in this study were evaluated. Based on the recognition ability of fruit wine quality, a judgment standard was established for a total of seven quality categories (color, transparency, sweetness, acidity, aroma, retention, and smoothness), and the quality of the above categories was scored (0–5 points): the larger the score, the stronger the characteristics. After assessing one sample, the panelists rinsed their mouths with pure water before tasting the next sample. Each sample of wine was evaluated twice, and the average value was calculated for further analysis.

### 4.9. Statistical Analysis

Microsoft Excel and Origin 2021 software were used for data processing and graphing. Statistical analysis was performed using SPSS (version 19.0). *p* < 0.05 indicates statistically significant difference amongst samples. Tukey’s HSD test was used for multiple comparisons. Origin 2018 was used for graphing.

## 5. Conclusions

While *S. cerevisiae*’s efficiency has long dominated winemaking, recent research reveals the untapped potential of non-*Saccharomyces* yeasts. When introduced early in fermentation alongside *S. cerevisiae*, these yeasts contribute unique flavor compounds, increasing a wine’s overall complexity and sensory appeal. The non-*Saccharomyces* strains *M. pulcherrima* and *K. apiculata* used in the present study were found to decrease alcohol content by 0.4–1.93%, and decrease total acidity by 18–24%. They also influenced the increase in number of identified volatile compounds in the wines. A particular advantage of wines made using non-*Saccharomyces* is the improvement in color intensity of citrus wines, a key factor influencing consumer preference. Meanwhile, this study observed a noteworthy decrease in off-notes associated with isobutanol (5–31% reduction) and octanoic acid (12–23% reduction) when co-fermenting with *M. pulcherrima* and *K. apiculata*, highlighting their potential to refine wine sensory characteristics. Moreover, mixed fermentation with *M. pulcherrima* and *K. apiculata* had a positive effect on the complexity of aroma by increasing volatile compounds, including ethyl octanoate, ethyl decanoate, and phenylethyl acetate, as well as ketones and terpenes. The presence of linalool and citronellol, introduced by *M. pulcherrima* and *K. apiculata*, further contributed to a floral aroma profile. Overall, the aroma retention, aroma, and sweetness qualities of the citrus wines were improved compared with Sc mono-inoculated fermented wines. In particular, the wines produced from *M. pulcherrima* with F15 mixed fermentation exhibited the best performance in terms of CI and the efficiency of sugar consumption, with the highest levels of esters, lowest levels of acids, and highest overall sensory evaluation scores, showing great potential in commercial citrus wine production.

## Figures and Tables

**Figure 1 molecules-29-01028-f001:**
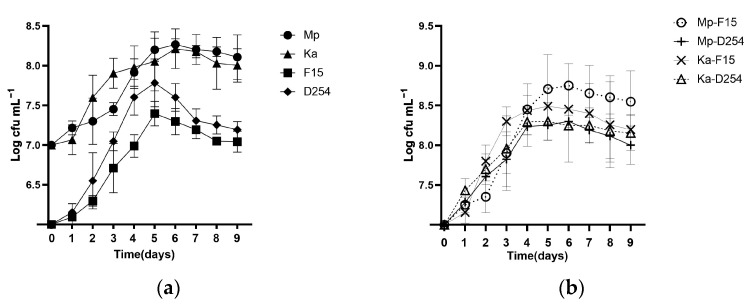
Growth kinetics of yeast population of (**a**) pure *M. pulcherrima* (Mp) culture, pure *K. apiculata* (Ka) culture, pure *S. cerevisiae* culture (F15 and D254), and (**b**) mixed culture of Mp and F15 (Mp-F15), Mp and D254 (Mp-D254), Ka and F15 (Ka-F15), and Ka and D254 (Ka-D254) in Chuanjian citrus juice.

**Figure 2 molecules-29-01028-f002:**
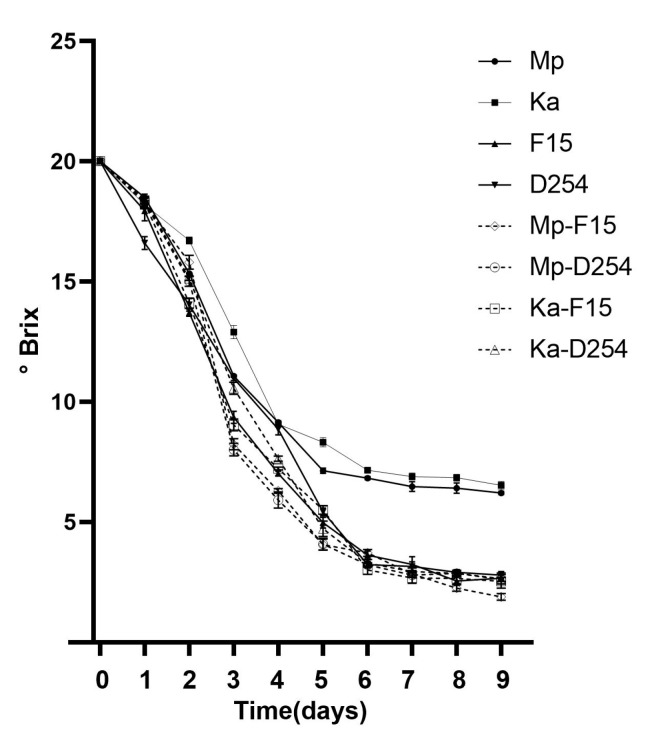
The sugar concentrations (°Brix) of pure NSc and Sc culture and mixed cultures during the citrus wine fermentation. Pure *M. pulcherrima* (Mp) culture, pure *K. apiculata* (Ka) culture, pure *S. cerevisiae* culture (F15 and D254), and mixed culture of Mp and F15 (Mp-F15), Mp and D254 (Mp-D254), Ka and F15 (Ka-F15), and Ka and D254 (Ka-D254).

**Figure 3 molecules-29-01028-f003:**
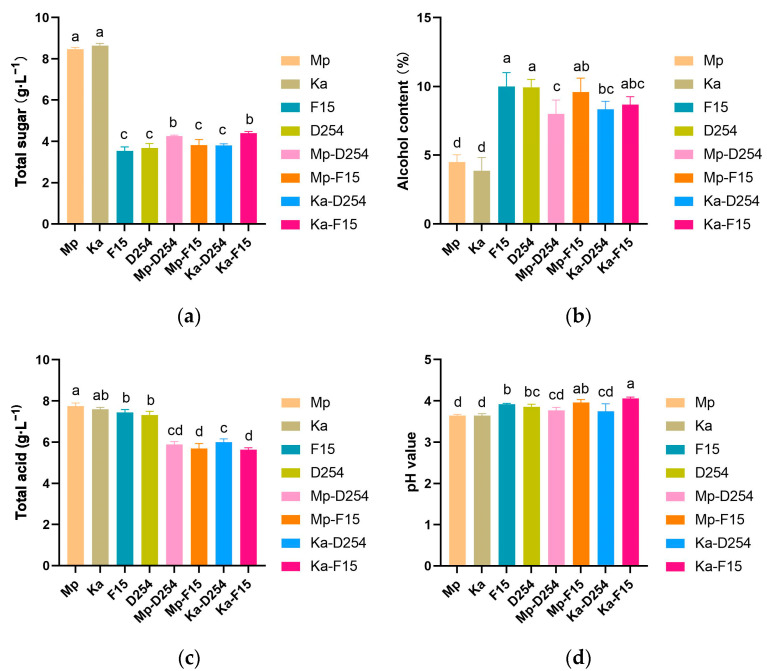
Total sugar (**a**), alcohol content (**b**), total acid (**c**), and pH value (**d**) in the pure-cultured and mixed-cultured Chuanjian citrus wine after fermentation. Different lower-case letters represent significant differences among groups of samples as tested by Tukey’s HSD test (*p* < 0.05).

**Figure 4 molecules-29-01028-f004:**
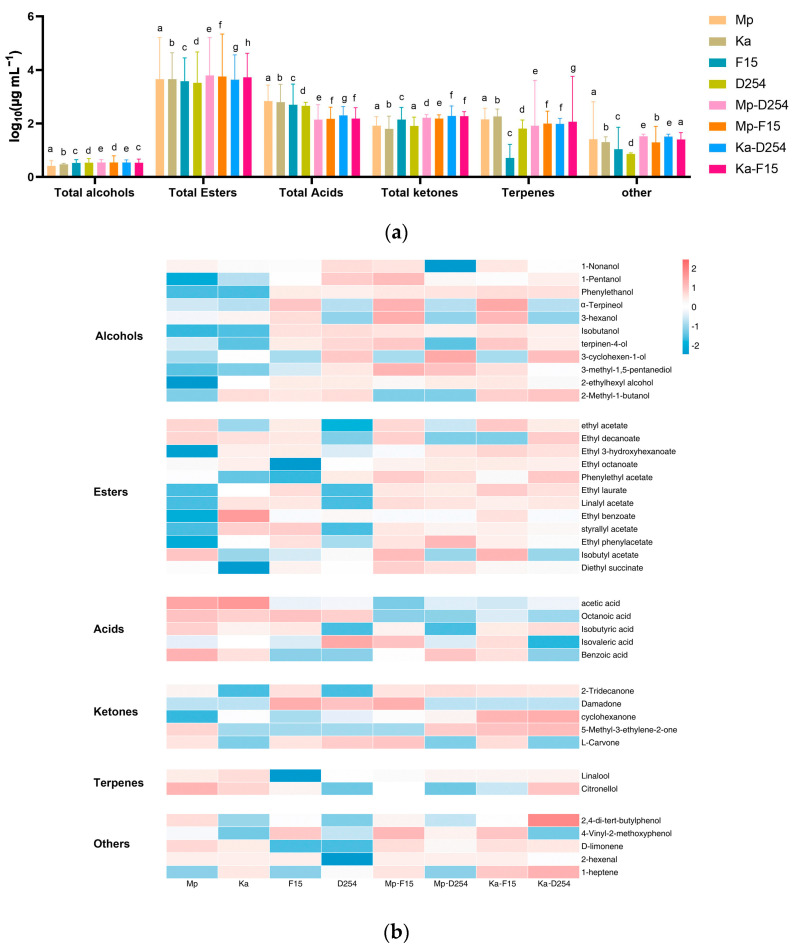
Aroma compound ratio (**a**) and heatmap of volatile compounds (**b**) in pure-cultured and mixed-cultured final Chuanjian citrus wines. Different lower-case letters represent significant differences among groups of samples as tested by Tukey’s HSD test (*p* < 0.05).

**Figure 5 molecules-29-01028-f005:**
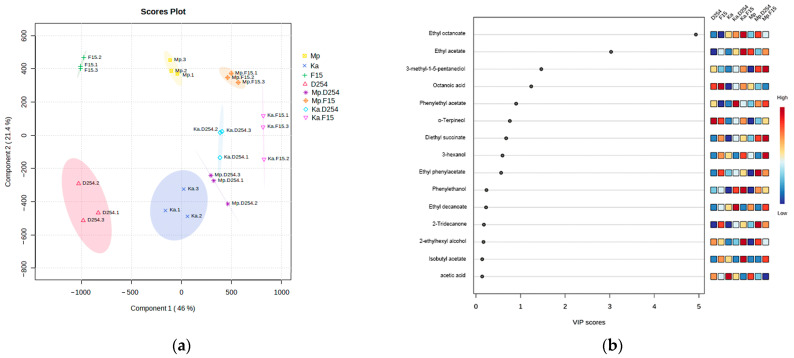
(**a**) PLS-DA plots based on the GC/MS data of citrus wine samples. The oval represents the 95% CI of the score calculated from each sample. (**b**) Variable Importance in Projection scores (VIP) of different groups.

**Figure 6 molecules-29-01028-f006:**
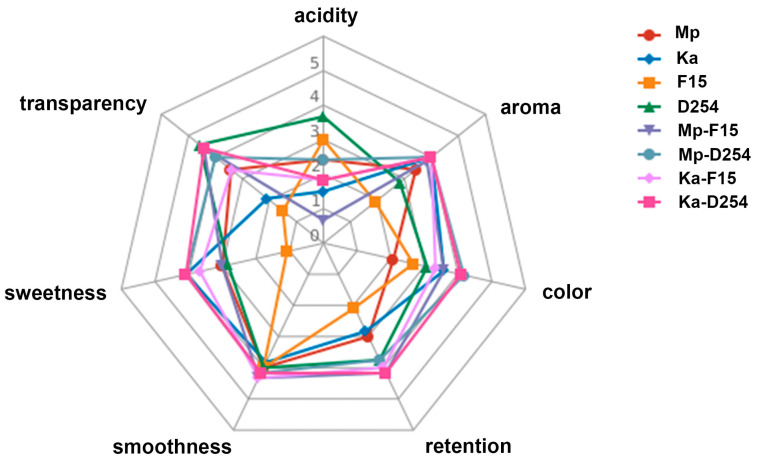
Sensory analysis of final Chuanjian citrus wines. Seven quality categories (color, transparency, sweetness, acidity, aroma, retention, and smoothness) were evaluated. The quality of the above categories is scored (0–5 points): the larger the score, the stronger the characteristics.

**Table 1 molecules-29-01028-t001:** Color parameters of Chunjian citrus wine.

Sample	A420 nm	A520 nm	A620 nm	CI	CT
Mp	0.31 ± 0.03 ^b^	0.11 ± 0.03 ^b^	0.05 ± 0.02 ^b^	0.47 ± 0.08 ^b^	3.00 ± 0.58 ^ab^
Ka	0.43 ± 0.15 ^b^	0.17 ± 0.12 ^b^	0.11 ± 0.07 ^ab^	0.70 ± 0.20 ^b^	3.71 ± 2.52 ^ab^
F15	0.29 ± 0.03 ^b^	0.08 ± 0.03 ^b^	0.04 ± 0.02 ^b^	0.41 ± 0.07 ^b^	4.16 ± 1.16 ^a^
D254	0.27 ± 0.04 ^b^	0.05 ± 0.01 ^b^	0.03 ± 0.01 ^b^	0.36 ± 0.04 ^b^	4.85 ± 0.81 ^a^
Mp-D254	0.39 ± 0.12 ^b^	0.12 ± 0.35 ^b^	0.14 ± 0.12 ^ab^	0.61 ± 0.15 ^b^	3.24 ± 0.21 ^ab^
Mp-F15	0.68 ± 0.11 ^a^	0.45 ± 0.04 ^a^	0.31 ± 0.33 ^a^	1.44 ± 0.25 ^a^	1.54 ± 0.36 ^b^
Ka-D254	0.51 ± 0.25 ^ab^	0.22 ± 0.18 ^b^	0.14 ± 0.12 ^ab^	0.88 ± 0.56 ^b^	2.90 ± 1.23 ^ab^
Ka-F15	0.42 ± 0.18 ^b^	0.15 ± 0.14 ^b^	0.10 ± 0.09 ^ab^	0.67 ± 0.38 ^b^	3.49 ± 1.46 ^ab^

Note: Different lower-case letters represent significant differences among groups of samples as tested by Tukey’s HSD test (*p* < 0.05); CI stands for color intensity; CT stands for color tonality.

## Data Availability

Data are contained within the article.

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
