# Peer review of "Effects of Non-Saccharomyces Yeasts and Their Pairwise Combinations in Co-Fermentation with *Saccharomyces cerevisiae* on the Quality of Chunjian Citrus Wine"

_molecules, 2024, doi:10.3390/molecules29051028_

Round 1

Reviewer 1 Report (Previous Reviewer 1)

Comments and Suggestions for Authors

The revised version of the manuscript has been improved according reviewer suggestion. However the some figures could be improved increasing the size of the letter or paste in the text with higher resolution.

Author Response

Response: Thank you for your valuable feedback on our figures. We have addressed your concerns by increasing the caption size and using higher resolution images in Figures 1-6. High-resolution graphics are available in a separate file upon request.

Reviewer 2 Report (New Reviewer)

Comments and Suggestions for Authors

Very interesting and innovative research on non-Saccharomyces yeast for production of distinct type of wines. 

My main criticism concerns the writing style in English. It's not so much about bad grammar as it is about style, and the problem lies in using language not typical of scientific writing, with many terms being misused. Many sentences, especially in the first part of the paper, don't make sense, presumably because the authors are using a literal translation from their native language to English, which doesn't work well. Having a proofreader review the paper won't be of much help; it needs to be reviewed by someone who is proficient in English and is knowledgeable in the field. I have also left comments within the paper itself at certain points.

Furthermore, the discussion needs to be expanded, with comparisons to other papers in the field. It's important to explain the phenomena behind the formation of certain compounds, what favors their formation, and so on. In the results section, all differences should be quantified so that it is clear and precise what has been decreased or increased by how much. As it's currently written, the paper is too general. The same applies to the conclusion.

Comments on the Quality of English Language

I already wrote about the quality of English 

Author Response

Response: We appreciate your valuable feedback on our manuscript. In response, we have thoroughly revised our writing, including enlisting a native speaker to review it. Inappropriate sentences have been rephrased or removed, and misused words have been corrected based on your specific points. We have addressed the issues you raised in lines 3, 12, 14, 16-17, 19-21 (abstract), 29-30, 31, 33, 34-35, 37-39, 60-64 (introduction), 67, 123, 152, 196-197 (results), 306 (discussion), 385, and 423 (methods). We believe these revisions have significantly improved the clarity and professionalism of our writing.

Meanwhile, the following changes have been made to the manuscript in response to your feedback: Expanded discussion section with systematic data analysis compared to previous studies (lines 235-334). Added explanations for compound formation and observed flavors with literature support (lines 285-288, 309-310, 321-322). Quantified all results (lines 152-158, 194-195). Relocated the paragraph about citrus wine (lines 44-53). Added standard deviations to Figure 1. Rewritten conclusion (lines 440-461). We believe these changes address your concerns and improve the overall quality of the manuscript.

Round 2

Reviewer 2 Report (New Reviewer)

Comments and Suggestions for Authors

The paper has been improved according to comments. 

Comments on the Quality of English Language

Even though the authors have corrected critical sentences and terms, the English still requires verification, particularly in terms of the meaning and style of the sentences.

This manuscript is a resubmission of an earlier submission. The following is a list of the peer review reports and author responses from that submission.

Round 1

Reviewer 1 Report

Comments and Suggestions for Authors

The present work studies the effect of simple or combined fermentation of saccharomycetic and non-saccharomycetic yeasts in citrus wine. The results may be relevant for innovation in this type of products, however, several aspects must be improved prior to publication.

Taking into account that citrus wines are less known in some parts of the world, I consider that more information about its industrial and economic development, the different varieties, as well as the regions where it is produced and consumed, should be added to the introduction to understand better the relevance of the present study.

Figure 5 is not legible, it must be improved.

The ANOVA test can be applied to figures 3 and 4a.

The expression of CFU/mL should be corrected in several sections of the text (for example lines: 300, 64, 66, 76). In particular, the CFU value of line 77 must be reviewed.

Check the subtitle of line 323.

In relation to color, in my opinion these values ​​are more relevant in the study of red grape wines. Explain in introduction, results and discussion, what type of color is expected to improve consumer perception and acceptance.

Add how these wines are currently produced and the future perspectives of this work in the large-scale production of this type of drink.

Add information about the yeasts used, origin and for what type of drinks they are marketed.

Author Response

Reviewer #1:

The present work studies the effect of simple or combined fermentation of saccharomycetic and non-saccharomycetic yeasts in citrus wine. The results may be relevant for innovation in this type of products, however, several aspects must be improved prior to publication.

Response: We appreciate your evaluation for our manuscript, and your valuable comments have been considered carefully and the corresponding content has been revised item by item as provided below.

Taking into account that citrus wines are less known in some parts of the world, I consider that more information about its industrial and economic development, the different varieties, as well as the regions where it is produced and consumed, should be added to the introduction to understand better the relevance of the present study.

Response: Thanks for your comments. Citrus fruits, such as orange, mandarin, and grapefruit, are used for wine and spirit production [1], forming varieties of citrus wines, including orange wine, lemon wine, mandarin wine, and blends (blended with other fruits or grapes). Citrus wines come in a spectrum of hues, from light gold to red, but a clear light amber with no sediment is generally preferred [2]. Citrus wine is widely consumed in the United States, Europe, and Asia, and the market share is growing rapidly globally [3]. Although citrus wine is not as familiar to customers as grape wine, however, the production of citrus wine has long history in Mediterranean and Asia, such as Italy's Limoncello, Spain's Arancello, and Japan's yuzu [4-6]. Meanwhile, the large production of citrus fruits, short harvest time and shelf life have led to the development of citrus wine production in China [7]. The information about citrus production regions, consuming places, industrial and economy development as well as the varieties has been included in the introduction of the manuscript in line 54-63.

Figure 5 is not legible, it must be improved.

Response: Thanks for your comments, Figure 5a and 5b have been replaced with higher resolution (300 dpi) images (Fig.5).

The ANOVA test can be applied to figures 3 and 4a.

Response: Thanks for your comment, ANOVA test has been applied to Figure 2 and 4a. Different lower-case letters represent significant differences among different groups of samples as tested by Tukey’s HSD test (P < 0.05), the according content were added in respective figure legends in line 122-124 and 161-162.

The expression of CFU/mL should be corrected in several sections of the text (for example lines: 300, 64, 66, 76). In particular, the CFU value of line 77 must be reviewed.

Response: Thanks for your careful review, the expression of CFU/mL in the according lines has been corrected. The format of index has been superscripted. See at line 74, 76, 78, 316, 323.

Check the subtitle of line 323.

Response: We are terribly sorry for the mistake, the subtitle of 4.6 has been corrected in line 346-347.

In relation to color, in my opinion these values ​​are more relevant in the study of red grape wines. Explain in introduction, results and discussion, what type of color is expected to improve consumer perception and acceptance.

Response: Thanks for your comments, depending on the type of citrus wine (pure cultured or mixed fermentation, types of yeast strains used, originally color of the citrus flesh), the color of citrus wines ranges from light gold to red. Generally, citrus wine with light amber in color having no visible precipitation is desired [2]. For Chuanjian citrus wine produced in this study, the color ranged from light gold to light amber. Our study showed that the mixed fermented citrus wine had higher color intensity and lower tonality values than Sc pure cultured wines (line 125-136). In our sensory evaluation, we found that the mixed fermentation citrus wine gained significantly higher value on the dimension of color than Sc pure fermented wine (line 210-214), which suggested that the amber color of citrus wine might be more fit in the consumers acceptance. The accordingly content has been added in introduction, results and discussion as you suggested in line 56-58, 134-135, 258-261.

Add how these wines are currently produced and the future perspectives of this work in the large-scale production of this type of drink.

Response: Thanks for your comment, the information of current producing process of citrus wine and future perspectives of our work in the industrious has been discussed in line 283-289.

Add information about the yeasts used, origin and for what type of drinks they are marketed.

Response: Thanks for your comment, the information of the yeasts used in this study were purchased from China Center of Industrial Culture Collection (CICC). Kloeckera apiculata CICC 1466 was originally isolated from Bretagne, French. Saccharomyces cerevisiae CICC 31964 has been widely used for wine production. Saccharomyces cerevisiae CICC 31898 was widely used for white wine production. Metschnikowia pulcherrima CICC 32343 was used for fruit wine production. The additional information of the four yeasts can be seen at http://sales.china-cicc.org/. The accordingly information has been added in line 303-309.

References

  1. Selli, S.; Canbas, A.; Varlet, V.; Kelebek, H.; Prost, C.; Serot, T. Characterization of the Most Odor-Active Volatiles of Orange Wine Made from a Turkish cv. Kozan ( Citrus sinensis L. Osbeck). Journal of Agricultural and Food Chemistry 2008, 56, 227-234, doi:10.1021/jf072231w.
  2. Bi, J.; Li, H.; Wang, H. Delayed bitterness of citrus wine is removed through the selection of fining agents and fining optimization. Frontiers in Chemistry 2019, 7, 185.
  3. Jagtap, U.B.; Bapat, V.A. Wines from fruits other than grapes: Current status and future prospectus. Food Bioscience 2015, 9, 80-96, doi:https://doi.org/10.1016/j.fbio.2014.12.002.
  4. Selli, S.; Canbas, A.; Varlet, V.; Kelebek, H.; Prost, C.; Serot, T. Characterization of the most odor-active volatiles of orange wine made from a Turkish cv. Kozan (Citrus sinensis L. Osbeck). Journal of agricultural and food chemistry 2008, 56, 227-234.
  5. Lee, J.-S.; Chang, C.-Y.; Yu, T.-H.; Lai, S.-T.; Lin, L.-Y. Studies on the quality and flavor of ponkan (Citrus poonensis hort.) wines fermented by different yeasts. Journal of Food and Drug Analysis 2013, 21, 301-309, doi:https://doi.org/10.1016/j.jfda.2013.07.004.
  6. Idise, O. Studies on wine production from orange (Citrus sinensis). Nigerian Journal of Science and Environment 2007, 10, 96-100.
  7. Liu, S.; Lou, Y.; Li, Y.; Zhang, J.; Li, P.; Yang, B.; Gu, Q. Review of phytochemical and nutritional characteristics and food applications of Citrus L. fruits. Frontiers in Nutrition 2022, 9, 968604.

Reviewer 2 Report

Comments and Suggestions for Authors

As it stands, this paper cannot be published. The subject is interesting, as is the methodology, but the paper itself is unfortunately still in draft form.

Many of the assertions in the introduction (such as the very first sentence) are not supported by bibliography. Similarly, the results contain general assertions that don't belong there, as in paragraph 2.2 between “The fermentation time” and “more sugars”. In the results you should only describe your observations and justify them with statistics. More most of your observation doesn't present clear numbers or statistics. At last the legend of the figures are mostly wrong: the legend of figure 2 is the legend of the figure 1 and the legend of the figure 3 is a global description of what should be a legend. 

Comments on the Quality of English Language

I don't have any comments about this part, the problems are not hear

Author Response

Reviewer #2:

As it stands, this paper cannot be published. The subject is interesting, as is the methodology, but the paper itself is unfortunately still in draft form.

Response: We appreciate your precious comment on our work. And by carefully studying your comments and reviewing our work, we have thoroughly improved our MS as following.

Many of the assertions in the introduction (such as the very first sentence) are not supported by bibliography.

Response: Thanks for your comment, the introduction has been checked thoroughly and all the statements has been supported by bibliography. Line 30, 32, 42.

 Similarly, the results contain general assertions that don't belong there, as in paragraph 2.2 between “The fermentation time” and “more sugars”. In the results you should only describe your observations and justify them with statistics. More most of your observation doesn't present clear numbers or statistics.

Response: Thanks for your comment, the general assertions have been removed and revised into statistical description. Line 77-78, 85-88, 95-102, 109-115, 117-120, 127, 129-132, 210-214, 216-218.

At last the legend of the figures are mostly wrong: the legend of figure 2 is the legend of the figure 1 and the legend of the figure 3 is a global description of what should be a legend. 

Response: We are terribly sorry for the mistakes we made, all the figure legends have been double checked carefully and revised accordingly at line 104-107 and 122-124.